# Corrosion Resistance of Novel Fly Ash-Based Forsterite-Spinel Refractory Ceramics

**DOI:** 10.3390/ma15041363

**Published:** 2022-02-12

**Authors:** Martin Nguyen, Radomír Sokolář

**Affiliations:** Faculty of Civil Engineering, Institute of Technology of Building Materials and Components, Brno University of Technology, Veveri 331/95, 602 00 Brno, Czech Republic; sokolar.r@fce.vutbr.cz

**Keywords:** forsterite, spinel, fly ash, corrosion resistance, refractory ceramics

## Abstract

This article aims to investigate the corrosion resistance of novel fly ash–based forsterite–spinel (Mg_2_SiO_4_-MgAl_2_O_4_) refractory ceramics to various corrosive media in comparison with reactive alumina–based ceramics. Because fly ash is produced in enormous quantities as a byproduct of coal-burning power stations, it could be utilized as an affordable source of aluminum oxide and silicon oxide. Corrosion resistance to iron, clinker, alumina, and copper was observed by scanning electron microscope with an elemental probe. The influence on the properties after firing was also investigated. Fly ash–based and reactive alumina–based mixtures were designed to contain 10%, 15% and 20% of spinel after firing. Raw material mixtures were sintered at 1550 °C for two hours. X-ray diffraction analysis and scanning electron microscopy were used to analyze sintered samples. The apparent porosity, bulk density, modulus of rupture, and refractory and thermo–mechanical properties were also investigated. The experimental results disclosed that the modulus of rupture, thermal shock resistance and microstructure were improved with increasing amounts of spinel in the fired samples. An analysis of the transition zones between corrosive media and ceramics revealed that all mixtures have good resistance against corrosion to iron, clinker, aluminum and copper.

## 1. Introduction

Refractory forsterite ceramics have played a significant role since the development of modern steelmaking technology. Due to the high melting point of forsterite refractory ceramics and their non-reactivity with iron at high temperatures, they have been predominantly implemented as a refractory lining of furnaces and regenerators in the metallurgical industrial sector. They have also been utilized in the cement and lime production industries as refractory lining for rotary kilns [1,2]. In the past decades, forsterite ceramics have also been utilized in electrotechnical engineering for ceramic-metal joints. Forsterite ceramics have relatively high coefficients of thermal expansion, which is comparable to the coefficient of metals used for joining [3].

In recent years, new ways of utilizing forsterite ceramics have emerged. Several researchers have investigated the sintering process of forsterite nanofibers with low thermal conductivity for their potential application as thermal insulation [4,5,6,7]. Other researchers are exploring the feasibility of a potential application of forsterite as a biomaterial in biomedicine for bone transplants due to its good compatibility with live tissue and high fracture toughness [8,9,10,11]. Researchers have also focused on the utilization of forsterite nanocrystals in the optical industry as a medium for optical lasers due to their great optical and mechanical properties [12,13].

Forsterite ceramics have a low thermal shock resistance due to their comparatively high coefficient of thermal expansion. However, this adverse effect of forsterite ceramics can be reduced by the incorporation of magnesium alumina spinel (MgO·Al_2_O_3_) directly into the raw material mixtures or indirectly by the addition of aluminum oxide as a raw material and its subsequent synthesis with magnesium oxide for the creation of spinel. Previous authors’ studies have explored the feasibility of synthesis of spinel from the addition of raw materials containing aluminum oxide into forsterite mixtures [14,15]. Refractory spinel ceramics are also widely adopted in various fields, predominantly as linings of various industrial kilns and furnaces due to their numerous advantages, i.e., a very high melting point of 2135 °C, a low coefficient of thermal expansion compared to coefficient of forsterite, better thermal shock resistance, and chemical and corrosion resistance [16,17,18,19,20].

Previous studies have proven that spinel incorporation into forsterite ceramics has been shown to improve physico-mechanical properties such as microstructure, mechanical properties and thermal shock resistance due to the embedment of spinel crystals that are located predominantly near the grain boundaries of larger forsterite crystals or bound in the amorphous matrix and filling the empty sections of cavities and pores in between [14,15,21].

Spinel refractory ceramics of industrial grade are commonly synthesized from alumina or bauxite combined with magnesium oxide [18,20,22]. However, fly ash, which is generated in vast amounts as a byproduct from coal-burning power stations all over the world, could be used as an affordable source of aluminum oxide and silicon oxide. Many researchers have focused on the implementation of fly ash into the mixture as a raw material for sintering predominantly aluminosilicate refractory ceramics [23,24]. Despite the increased attention on the implementation of fly ash in the synthesis of aluminosilicate refractories, fly ash has rarely been looked at as the research object in the synthesis of other ceramic refractories that include silicon oxide and/or aluminum oxide, for example, forsterite–spinel refractory ceramics.

Corrosion resistance of the refractory ceramics is an important characteristic of every refractory product made. It is the ability of the tested ceramic material to withstand deterioration against the corrosive substance. Therefore, corrosion resistance to various corrosive media is a key characteristic of every refractory ceramic. It is known from previous research and the literature that different refractories have different corrosion resistance based on the chemical composition of the refractory and the corrosive materials. Acidic refractories such as silica, zirconia and aluminosilicate refractories are generally highly resistant to acidic materials but have low corrosion resistance to basic materials with MgO or CaO in their chemical composition [1,2,25,26].

On the contrary, basic refractories are highly resistant to basic materials such as clinker, lime, basic slags and alkaline materials with lower corrosion resistance to acidic materials. Moreover, they are stable, with high corrosion resistance to various metals and their slags. Many researchers have focused on testing the corrosion resistance of refractory ceramics to different corrosive media on magnesia or spinel refractory ceramics with positive results [25,26,27,28].

The main objective of this research is to investigate the influence of corrosion resistance by various corrosive media on fly ash–based forsterite–spinel ceramics in comparison with reactive alumina–based ceramics. The corrosion resistance of all designed forsterite–spinel mixtures was evaluated by examining the transition zone between corrosive media and ceramics using a scanning electron microscope with an elemental probe. In addition, the phase composition of fired forsterite–spinel ceramics was analyzed through X-ray diffraction analysis. The microstructure of fired test samples was further observed using scanning electron microscopy, and the physico–mechanical, refractory and thermo–mechanical properties of forsterite–spinel ceramics were also evaluated.

## 2. Materials and Methods

### 2.1. Raw Materials

Calcined caustic magnesite (CCM) was obtained from Magnesite Works (Jelsava, Slovakia), olivine from Norway, talc from Fichema (Brno, Czech Republic), coal fly ash from Mělník power plant (Mělník, Czech Republic), reactive alumina from Almatis (Ludwigshafen, Germany) and kaolin from Sedlecký kaolin (Božičany, Czech Republic). The chemical composition of the involved raw materials is presented in Table 1. Chemical analysis and X-ray fluorescence was used to determine the chemical composition.

The coarser raw materials were pre-treated in the vibration mill to achieve a particle size range of 1–64 µm (where d_50_ = 10–30 µm). Particle size distribution was determined by laser granulometer (Malvern Mastersizer, Malvern Panalytical, Malvern, UK). The raw materials were then subjected to determination of mineralogical composition by means of X-ray diffraction analysis (XRD; Panalytical Empyrean, Panalytical B.V., Almelo, Netherlands). with CuKα as a radiation source, an accelerating voltage of 45 kV and a beam current of 40 mA. Olivine, fly ash and alumina were also subjected to determination of morphology in a scanning electron microscope (Tescan MIRA3, Tescan Orsay Holding a.s., Brno, Czech Republic).

Figure 1 represents the mineralogical composition of the raw materials. The major crystalline phase in CCM was periclase (MgO), with trace amounts of iron compounds. Forsterite (2MgO·SiO_2_) was the major crystalline phase in olivine, with minor crystalline phases of fayalite (2FeO·SiO_2_), serpentinite (3MgO·2SiO_2_·2H_2_O) and quartz (SiO_2_). The only crystalline phase in talc powder was talc (3MgO·4SiO_2_·H_2_O). The crystalline phases in coal fly ash were mullite (3Al_2_O_3_·2SiO_2_) and quartz (SiO_2_). The only crystalline phase in reactive alumina was corundum (Al_2_O_3_). Kaolin was primarily composed of kaolinite (Al_2_O_3_·2SiO_2_·2H_2_O) and traces of biotite (K(Mg,Fe)_3_AlSi_3_O_10_(OH)_2_). The existence of an amorphous glass phase is indicated by the background curvature in olivine and fly ash diffractograms, and the background scattering (noise) is caused by the presence of iron compounds due to the interference with a CuKα radiation source.

The scanning electron microscope (SEM) microphotographs of all untreated raw materials are presented in Figure 2a–f. The particles in CCM agglomerated together, with a particle size in a range of 5–50 µm. Untreated olivine has an apparent fibrous microstructure. Particles in talc have a foliated or fibrous microstructure, with different size flakes. The reactive alumina is composed of fine particles with d_50_ = 1.9 µm and that tend to cluster together. The untreated coal fly ash is mainly composed of spherical particles ranging in size, with a diameter of 0.4–90 µm (d_50_ = 14 µm). Kaolin platelets are clustered together in a general sheet appearance.

### 2.2. Sample Preparation

In this work, six different mixtures were designed. Coal fly ash was the source of Al_2_O_3_ in mixtures FA-S10, FA-S15 and FA-S20, and reactive alumina was the source of Al_2_O_3_ in mixtures RA-S10, RA-S15 and RA-S20. The different sources of aluminum oxide were selected for spinel synthesis and comparison of properties of the final forsterite–spinel ceramics. The designations S10, S15 and S20 correspond to the theoretical amount of spinel in the mixture after synthesis. The mixture composition from raw materials is presented in Table 2.

First, the raw materials were accurately weighed, mixed and homogenized in a container by means of a rotary mechanical homogenizer for 24 h. After homogenization, mixtures were then mixed with a varying amount of water, utilizing a Pfefferkorn deformation apparatus (standard ČSN 72 1074) to achieve the optimal plasticity. The optimal plasticity (P_opt_) was achieved when the ratio of sample height after deformation (h_def_) to sample height before deformation (h_0_) was equal to 0.6, as defined in Equation (1).
P_opt_ = h_def_/h_0_.(1)

Test samples from all mixtures for all tests were obtained by molding plastic paste into the brass molds. The green samples were then dried until a constant weight in a laboratory drier at 105 °C. The dried samples were fired at 1550 °C with a heating rate of 4 K/min in a laboratory furnace with an air atmosphere. The soaking time was two hours at maximum temperature.

### 2.3. Characterization

The dimensions of the test samples were 20 × 25 × 100 mm^3^ for the measurement of a change in dimension during firing (standard EN 993-10:1997) and modulus of rupture (MOR; Testometric M350-20CT, Testometric Co. Ltd., Rochdale, UK), according to the standard EN 993-6:1995. A vacuum water absorption method with subsequent hydrostatic weighing (standard EN 993-1:1995) was used to determine apparent porosity, water absorption and bulk density on the same test samples. Refractoriness (standard EN 993-12:1997) of the mixtures was tested on a set of three pyrometric cones that were prepared according to the standard EN 993-13:1995. The refractoriness was performed in a small laboratory furnace with an observation port equipped with a digital camera that allowed real-time observation of the furnace and of the pyrometric cones.

On cylindrical test samples with a height of 50 mm and 50 mm in diameter, refractoriness under load (standard ISO 1893:2007) was investigated, and the temperature at 0.5% deformation (T_0.5_) was evaluated. Thermal shock resistance (standard EN 993-11:2007; method B) was determined by a parameter *residual MOR (MOR_res_)* in percent, which is a ratio between the MOR of thermally cycled samples (MOR_cyc_) and samples at a laboratory temperature (MOR), according to Equation (2). A residual MOR parameter enables a quantitative approach for measuring the thermal shock resistance. Test samples for thermal shock resistance were prisms with dimensions of 230 × 64 × 54 mm^3^.
MOR_res_ = (MOR_cyc_/MOR)×100(2)

X-ray diffraction analysis was also performed on test samples to determine their mineralogical composition. Fluorite (CaF_2_) was added to the samples as an inert standard for the quantitative analysis of all samples. 

### 2.4. Testing of Corrosion Resistance of Forsterite–Spinel Ceramic Samples

Corrosion resistance (standard CEN/TS 15418:2006) was tested using the crucible method, with prism-shaped crucibles made and sintered from the same plastic paste for all six designed mixtures. The crucibles were made with utilization of a two-part brass mold with a prism base of 110 × 110 × 84 mm^3^ and the top cylindrical part with a diameter of 60 mm and height of 60 mm. The dimensions of the crucibles after firing were due to the firing shrinkage of approximately 100 × 100 × 76 mm^3^, with hollow cylindrical centers with a 55 mm diameter and 55 mm depth, which is in conformity with the dimensions specified in the standard CEN/TS 15418:2006. 

The corrosive media used were iron, clinker, copper and aluminum to test the endurance of the designed forsterite–spinel mixtures against the corrosion of the molten materials. According to the literature [1,2], the corrosion resistance of the industrially produced forsterite ceramics is good against all used corrosive media mentioned above. The firing temperature for the corrosion resistance test was set at the melting point of the individual corrosive media—1538 °C for iron, 1450 °C for clinker, 1085 °C for copper and 660 °C for aluminum—to simulate the exposure of the refractory lining inside the kiln or furnace. The heating rate for the corrosion resistance test was 5 K/min, and soaking time was five hours at the maximum temperature, as specified by the standard CEN/TS 15418:2006. 

The results of the corrosion resistance of all six mixtures were evaluated by SEM analysis with a secondary electron (SE) detector and a back-scattered electron (BSE) detector to analyze the microstructure and phase transition zones at the area of contact between the ceramics and corrosive media. An energy dispersive X-ray spectroscopy (EDX) probe was also utilized to determine the elemental analysis of the areas near the transition zone.

## 3. Results and Discussion

### 3.1. Mineralogical Composition and Microstructure

The XRD diffractograms of all mixtures are presented in Figure 3. The major crystalline phase for all mixtures is forsterite (2MgO·SiO_2_), with minor crystalline phases of spinel (MgO·Al_2_O_3_), periclase (MgO) and monticellite (CaO·MgO·SiO_2_) minerals. Fluorite (CaF_2_) was added to the samples for the quantitative phase determination. The curved background of the XRD diffractograms signifies the presence of an amorphous glass phase, and the background scattering indicates the presence of iron compounds due to the use of CuKα as a radiation source. The quantitative analysis of the phase composition is presented in Table 3.

As can be seen in Figure 3, unreacted Al_2_O_3_ was not detected in the fired samples, and at the same time, no traces of mullite were detected. Therefore, the mullite in fly ash (Figure 1) had completely decomposed and recrystallized with magnesium oxide into spinel. It can therefore be concluded that all aluminum oxide reacted and formed spinel. Unreacted periclase was observed in the XRD diffractogram due to the presence of an amorphous glass phase, which inhibited periclase’s reaction in forming forsterite. However, forsterite, spinel and periclase have excellent refractory properties. The amount of amorphous phase was higher in samples with fly ash due to its higher content in fly ash and due to the presence of flux oxides. All samples also contained a minor amount of monticellite, which formed due to the presence of calcium oxide in the raw materials. The theoretical value of formed spinel in all mixtures correlated with the results of quantitative analysis.

Figure 4a–f represents the SEM microphotographs of the morphology and microstructure of the samples with fly ash (FA-S10, FA-S15, FA-S20) fired at 1550 °C for two hours. It can be observed in Figure 4d–f that the tetragonal dipyramidal spinel crystals formed in clusters with diameters of 2–4 µm. They were located predominantly at the grain boundaries of larger forsterite crystals or bound in the amorphous matrix, filling some sections of the cavities and pores in between (see Figure 4a–c).

Spinel crystals that were synthesized from the mixtures with fly ash (Figure 4d–f) were less uniform and more irregular, with indications of polymorphism and crystal deformations. The crystal deformations can be observed in Figure 4d,e, and the polymorphic crystallization can be observed in the bottom of Figure 4f. This can be explained by the fact that spinel crystals formed indirectly from the decomposition of mullite and the subsequent reaction with magnesium oxide in mixtures with fly ash.

Figure 5a–f represents the SEM microphotographs of the morphology and microstructure of the samples with reactive alumina (RA-S10, RA-S15, RA-S20) fired at 1550 °C for two hours. It can be observed in Figure 5d–f that the tetragonal dipyramidal spinel crystals formed in clusters with diameters of 2–6 µm. With increasing content of reactive alumina in the mixture, the spinel crystal size was also increased. They were located primarily at the grain boundaries of larger forsterite crystals or bound in the amorphous matrix and filling some sections of the cavities and pores in between (Figure 5a–c).

The spinel crystals that were synthesized from the mixtures with reactive alumina were more uniform, and the crystallization was complete, with smooth surfaces and without any indication of polymorphism or deformations. This could be attributed to spinel crystals that crystallized directly from reactive alumina and magnesium oxide without any intermediate phase.

### 3.2. Physico-Mechanical Properties

Table 4 presents the results of the experiments that utilized the vacuum water absorption method, together with hydrostatic weighing to determine the apparent porosity, water absorption and bulk density, as well as the results of the MOR of fired samples from all six designed mixtures.

Apparent porosity and water absorption decreased with increased amounts of formed spinel in the structure in both fly ash–based mixtures and reactive alumina–based mixtures. This can be attributed to the higher firing shrinkage with increasing amounts of fly ash and reactive alumina in the mixtures for the subsequent spinel synthesis. The higher firing shrinkage resulted in higher densification of samples, which led to a decrease in apparent porosity and water absorption. The higher decrease in porosity and water absorption in fly ash–based mixtures is due to the creation of a more amorphous glass phase, resulting from increased amounts of flux oxides due to the increased content of fly ash in mixtures FA-S15 and FA-S20 for the subsequent spinel synthesis. Apparent porosity was higher in fly ash–based mixtures than alumina–based mixtures due to the expansion stage at 1250 °C, which was caused by the reaction of flux oxides and amorphous phase. This phenomenon is described in more detail in [15].

A higher bulk density of reactive alumina mixtures was caused by the denser structures of these samples due to the lower porosity and utilization of larger quantities of raw materials with higher bulk densities. The highest MOR in fly ash mixtures was achieved in mixture FA-S15, with a MOR value of 18.4 MPa. Similarly, the highest MOR in reactive alumina mixtures was achieved by the RA-15 mixture, with an MOR value of 22.6 MPa. The decrease of MOR in fly ash mixture FA-S20 is attributed to the large quantity of flux oxides from fly ash. Comparably, the decrease of MOR in reactive alumina mixture RA-S20 was caused by the increased amount of amorphous phase.

When the quantity of synthesized spinel increased in the mixture, the apparent porosity and water absorption decreased, while the bulk density and MOR increased. It can be concluded that increasing the quantity of synthesized spinel in forsterite ceramics leads to improved physico-mechanical properties.

### 3.3. Refractory and Thermo-Mechanical Parameters

Table 5 contains the results of firing shrinkage, refractoriness, refractoriness under load and residual MOR, which is a parameter for the determination of thermal shock resistance for all six designed mixtures.

As can be seen in Table 5, the firing shrinkage increased in mixtures with fly ash due to the elevated volume of flux oxides, which promote sintering. Firing shrinkage was higher in fly ash–based mixtures as opposed to reactive alumina–based mixtures, in which the spinel crystallized from mullite and magnesium oxide. Consequently, spinel has higher density than mullite, which also promoted shrinkage during firing in fly ash–based mixtures.

Larger quantities of flux oxides in fly ash mixtures also caused lower refractoriness and refractoriness under load of these mixtures, as opposed to mixtures with reactive alumina. However, the maximum impairment caused only a 5% decrease in the refractoriness of S20 mixtures and a 7% decrease in refractoriness under load of the S15 mixtures. Residual MOR is a parameter of thermal shock resistance. The highest values of residual MOR of both fly ash–based and reactive alumina–based mixtures had samples with 15% spinel (S15) after firing.

The mixtures with 20% spinel (S20) contained larger quantities of the amorphous phase, which caused marginally lower values of refractoriness, refractoriness under load and residual MOR. In general, the addition of a small quantity of spinel into the forsterite ceramics leads to better MOR and thermal shock resistance due to the improvement in microstructure. In addition, spinel ceramics have higher thermal shock resistance due to the lower value of coefficient of thermal expansion compared to forsterite [3,19].

### 3.4. Corrosion Resistance of Forsterite–Spinel Ceramics

Section 3.4.1, Section 3.4.2, Section 3.4.3 and Section 3.4.4. contain the results of the corrosion resistance of forsterite–spinel ceramics, prepared from all six designed mixtures with utilization of the SEM microphotographs of the transition zones between the corrosive media and the ceramics. The resulting resistance against the corrosion by the corrosive media is evaluated by the overall microstructure near the transition zone between the ceramics and corrosive media and the depth of penetration of the corrosive media into the ceramics. The corrosion resistance of forsterite–spinel ceramics was tested by molten iron, copper, aluminum and clinker.

#### 3.4.1. Corrosion Resistance of Ceramics to Iron

The microphotographs from SEM of the transition zone between iron and forsterite–spinel ceramics of all designed mixtures are presented in Figure 6 and Figure 7. Due to the heavier atomic number of iron, the BSE detector can be utilized to detect various levels of signal and differentiate between iron (light grey) and ceramics (dark grey).

Figure 6a–f represents the corrosion resistance to iron of samples of fly ash–based mixtures FA-S10, FA-S15 and FA-S20. These samples had an increased porosity in the proximity of the transition zone between iron and ceramics, which was caused by the creation and formation of fayalite (2FeO·SiO_2_) from iron oxide and an equimolar amount of amorphous silica, according to Equation (3).
2FeO + SiO_2_ → 2FeO·SiO_2_.(3)

The increase in porosity, which can be seen in Figure 6a–c, was stronger in fly ash–based mixtures, as the amount of the amorphous phase was higher than that of reactive alumina–based mixtures. The higher ratio of flux oxides and mullite decomposition also led to the formation of additional amorphous phase. Due to the higher porosity in the proximity of the transition zone, the EDX probe also detected olivine with a higher concentration of iron oxide, which diffused more easily into the pore structure. Olivine is a solid solution between forsterite (2MgO·SiO_2_) and fayalite (2FeO·SiO_2_), with a general formula of (Mg^2+^, Fe^2+^)_2_SiO_4_. The olivine found in the sample contained 5–10% of fayalite, with up to 30% in the proximity of the transition zone. The depth of penetration of iron into the fly ash–based mixtures was 1–2 mm, with lower values for mixtures with increased content of spinel (FA-S15, FA-S20).

Figure 7a–f represents the corrosion resistance to iron of samples of reactive alumina–based mixtures RA-S10, RA-S15 and RA-S20. These samples also had weakly increased porosity (Figure 7a–c) in the proximity of the transition zone between iron and ceramics, which was caused by the creation and formation of fayalite. However, the increase in porosity was lower than that of fly ash–based mixtures. This also allowed the diffusion of iron into the porous ceramic structure, which is clearly visible in Figure 7d–f.

However, due to the low solubility of spinel to iron oxide [29], the increased amount of spinel in the forsterite ceramics (mixtures S15, S20) led to more distinct transition zones between iron and ceramics with larger grains of forsterite–spinel matrix (dark grey) and iron oxide-fayalite (light grey). As a result, the corrosion resistance of forsterite–spinel ceramics to molten iron was negligible, with a depth of penetration of iron into the ceramics of 0–2 mm in reactive alumina–based mixtures. The depth of penetration was lower with increased content of spinel in the mixtures.

#### 3.4.2. Corrosion Resistance of Ceramics to Clinker

The microphotographs from SEM of the transition zone between clinker and forsterite–spinel ceramics of all prepared mixtures are presented in Figure 8 and Figure 9. Due to the similar atomic numbers of clinker compounds and ceramics, the BSE detector could not be utilized to differentiate between clinker and ceramics. Therefore, a secondary electron (SE) detector with an EDX probe and larger magnification was used. Figure 8a–c and Figure 9a–c illustrate the microstructure near the transition zone between the clinker and ceramics. Figure 8d–f and Figure 9d–f illustrate the microstructure with a larger magnification.

Figure 8a–f represents the corrosion resistance to clinker of fly ash–based mixtures FA-S10, FA-S15 and FA-S20. The clinker reacted with ceramics at the transition zone and caused additional formation of monticellite (CaO·MgO·SiO_2_) from forsterite and dicalcium silicate, according to Equation (4), and the creation of merwinite (3CaO·MgO·2SiO_2_) from one mole of forsterite and three moles of dicalcium silicate, according to Equation (5). Both minerals were identified by the EDX probe.
2MgO·SiO_2_ + 2CaO·SiO_2_ → 2(CaO·MgO·SiO_2_),(4)2MgO·SiO_2_ + 3(2CaO·SiO_2_) → 2(3CaO·MgO·2SiO_2_).(5)

Due to the slow cooling of the samples, tricalcium silicate dissolved into dicalcium silicate and lime (CaO), according to Equation (6) [30]. Subsequently, lime reacted with spinel in the proximity of the transition zone to form tricalcium aluminate, with simultaneous precipitation of periclase (MgO), according to Equation (7). Isometric hexoctahedral periclase crystals are visible in Figure 8d–e, as small cube-shaped crystals with diameters of 1–2 µm were scattered on the dicalcium aluminate/amorphous silica melt.
3CaO·SiO_2_ → 2CaO·SiO_2_ + CaO,(6)
3CaO + MgO·Al_2_O_3_ → 3CaO·Al_2_O_3_ + MgO.(7)

Figure 8f represents the spinel crystals in the amorphous phase located in the proximity of the transition zone. The corrosion resistance of fly ash–based mixtures to clinker was minimal, with the depth of penetration of 2–3 mm. With the increased content of spinel in the mixtures (FA-S15, FA-S20), the depth of penetration was lower than in mixture FA-S10.

Figure 9a–f represents the corrosion resistance to clinker of reactive alumina–based mixtures RA-S10, RA-S15 and RA-S20. Figure 9a–c represents the overall microstructure in the proximity of the transition zone, and Figure 9d–f represents the microstructure with a higher magnification of 5000×. Periclase crystals are clearly visible in Figure 9d–f as small white cube-shaped crystals with diameters of 1–2 µm scattered on the dicalcium aluminate/amorphous silica melt (darker grey). The corrosion resistance of reactive alumina–based mixtures to clinker was minimal, with better results than fly ash–based mixtures. The depth of penetration of clinker into the reactive alumina–based mixtures was 1–2.5 mm. With the increased content of spinel in the mixtures (RA-S15, RA-S20), the depth of penetration of clinker was lower than in mixture RA-S10.

#### 3.4.3. Corrosion Resistance of Ceramics to Aluminum

The microphotographs from SEM of the transition zone between aluminum and forsterite–spinel ceramics synthesized from all mixtures are presented in Figure 10 and Figure 11. The SE detector could clearly distinguish between the aluminum metal (darker and smooth) and ceramics (lighter and porous). The transition zone is clearly visible in the center of Figure 10a–c and Figure 11a–c. Between the aluminum metal and the ceramics is a darker smooth layer, visible in the bottom of Figure 10d–f and Figure 11d–f. The EDX probe detected that this darker layer was composed mainly of melted aluminum, which filled the outer pores of the ceramics, and aluminum oxide (Al_2_O_3_).

Figure 10a–f illustrates the corrosion resistance to aluminum of fly ash–based mixtures FA-S10, FA-S15 and FA-S20. The corrosion resistance of fly ash–based mixtures to aluminum metal was very good, with the depth of penetration of aluminum around 1 mm for fly ash–based mixtures. The aluminum oxide layer (dark grey) was thinner in mixture FA-S20 due to the higher content of spinel, which also led to better corrosion resistance of this mixture.

Figure 11a–f illustrates the corrosion resistance to aluminum of reactive alumina–based mixtures RA-S10, RA-S15 and RA-S20. The darker grey layer in the middle of Figure 11a,b and in the bottom of 11c represents the layer of oxidized aluminum. The corrosion resistance of reactive alumina–based mixtures to aluminum metal was very good, with a depth of penetration of aluminum of 0.2–0.6 mm for reactive alumina–based mixtures. The layer of oxidized aluminum (dark grey) was very thin in mixtures RA-S15 and RA-S20 due to the higher content of spinel, which also led to better corrosion resistance of these mixtures.

#### 3.4.4. Corrosion Resistance of Ceramics to Copper

The microphotographs from SEM of the transition zone between copper and forsterite–spinel ceramics are presented in Figure 12 and Figure 13. The SE detector could clearly distinguish between the copper metal (lighter and smooth) and ceramics (darker and porous). The transition zone between copper and ceramics is clearly visible in the middle of Figure 12a–c and Figure 13a–c. Between the copper metal and the ceramics is a very thin layer of a few micrometers that is darker than the copper metal and lighter than ceramics (middle of Figure 12d–f). The EDX probe detected that this thin layer was composed primarily of melted copper, which filled the pores of the ceramics, and part of the copper that oxidized into copper oxide (CuO). The bottom halves of Figure 12a–c contain almost pure copper, with small quantities of copper oxide.

The corrosion resistance of fly ash–based mixtures FA-S10, FA-S15 and FA-S20 to copper metal was excellent, with a depth of penetration of 0.01–0.013 mm. All three fly ash–based mixtures have very high corrosion resistance to molten copper.

The transition zone between copper and forsterite–spinel ceramics is clearly visible in the middle of Figure 13a–c. Between the copper metal and the ceramics is a very thin layer of a few micrometers that is darker than the copper metal and lighter than the ceramics, visible in the middle of Figure 13d–f. The corrosion resistance of reactive alumina–based mixtures RA-S10, RA-S15 and RA-S20 to copper metal was excellent, with a depth of penetration of 0.003–0.008 mm. All three reactive alumina–based mixtures have very high corrosion resistance to molten copper.

#### 3.4.5. Discussion of Corrosion Resistance Results

According to the literature, the corrosion resistance of forsterite ceramics is very good in response to the corrosive effect of molten iron or iron slags. Similarly, clinker does not react with forsterite up to 1500 °C; therefore, the corrosion resistance of forsterite to clinker is also very high [1,2,31,32]. This is also consistent with the results of corrosion resistance to iron and clinker in this paper. The corrosion resistance to iron and clinker of both fly ash–based and reactive alumina–based mixtures was very high. With increased content of spinel in the mixtures (FA-S15, RA-S15, FA-S20 and RA-S20), the corrosion resistance was even better. This was due to the increased content of Al_2_O_3_, which has a low solubility to iron oxide [26,28,29,31].

The high content of MgO (in forsterite and periclase) in all mixtures also results in very good corrosion resistance to non-ferrous metals such as aluminum and copper due to the excellent oxidation resistance of MgO through solid solution phase formation [29,31]. The aluminum and copper metals that are in contact with MgO-based refractories such as forsterite form a protective oxide layer on the surface of the ceramics (transition zone), which inhibits any further corrosion of the ceramics. This is also in agreement with the results of the corrosion resistance of forsterite–spinel ceramics to aluminum and copper in this paper. The corrosion resistance was slightly improved in mixtures with reactive alumina (RA-S10, RA-S15, RA-S20) as opposed to fly ash–based mixtures (FA-S10, FA-S15, FA-S20) due to the lower content of amorphous phase and flux oxides.

## 4. Conclusions

Refractory forsterite–spinel ceramics were successfully sintered from fly ash–based and reactive alumina–based raw materials to compare the resulting properties after the firing of fly ash–based mixtures (FA-S10, FA-S15, FA-S20) compared to reactive alumina–based mixtures (RA-S10, RA-S15, RA-S20). The corrosion resistance of all six ceramic mixtures to iron, clinker, aluminum and copper was also tested. The increased content of spinel in the forsterite–spinel ceramics led to improved physico–mechanical properties such as MOR and thermal shock resistance, especially in mixtures FA-S15 and RA-S15 with 15% spinel and mixtures FA-S20 and RA-S20 with 20% spinel. The spinel crystals in reactive alumina–based mixtures that formed from alumina and magnesium oxide were more uniform and without cracks, and the resulting properties, such as MOR and thermal shock resistance, improved with increased alumina (RA-S15, RA-S20) content in the mixture, without impairing refractory properties. The spinel crystals in fly ash–based mixtures that formed from mullite decomposition in the presence of magnesium oxide in mixtures FA-S10, FA-S15 and FA-S20 were less uniform and had cracks, but the resulting properties (MOR, thermal shock resistance) were improved in mixture FA-S15 compared to mixtures FA-S10 and FA-S20, with minor impairments to the refractory properties in comparison with alumina–based mixtures. Mixtures FA-S15 and RA-S15, containing 15% spinel after firing, had the best resulting properties of all designed mixtures.

A microstructural analysis by SEM of the transition zones between the corrosive media and forsterite–spinel ceramics revealed that all mixtures had good resistance against corrosion from iron, clinker, aluminum and copper. The highest corrosion resistance to all tested corrosive media was for mixtures FA-S15, RA-S15 and RA-S20, with 15% and 20% spinel. The depth of penetration of iron was 0–2 mm in all mixtures. In corrosion from clinker, the depth of penetration was 1–2 mm in alumina–based mixtures (RA-S10, RA-S15 and RA-S20) and 2–3 mm in fly ash–based mixtures (FA-S10, FA-S15 and FA-S20). In corrosion from aluminum metal, the depth of corrosion was 0.5–1 mm and only 0.005–0.01 mm in corrosion by copper metal in all tested mixtures.

In conclusion, mixtures FA-S15 and RA-S15, with 15% spinel in forsterite ceramics, improved the microstructure, MOR and thermal shock resistance while retaining good refractory properties. Corrosion resistance to all tested corrosive media was also very promising.

## Figures and Tables

**Figure 1 materials-15-01363-f001:**
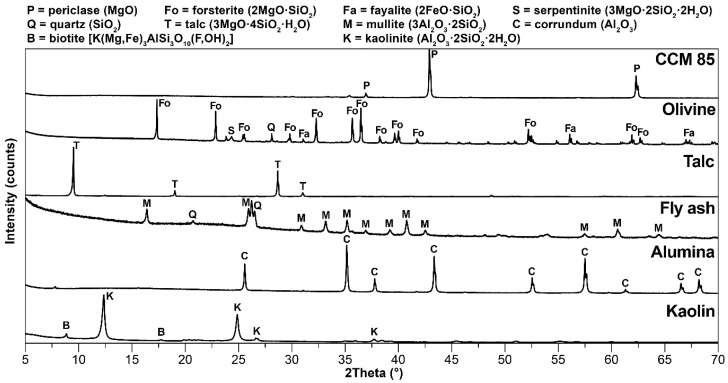
X-ray diffraction analysis of raw materials.

**Figure 2 materials-15-01363-f002:**
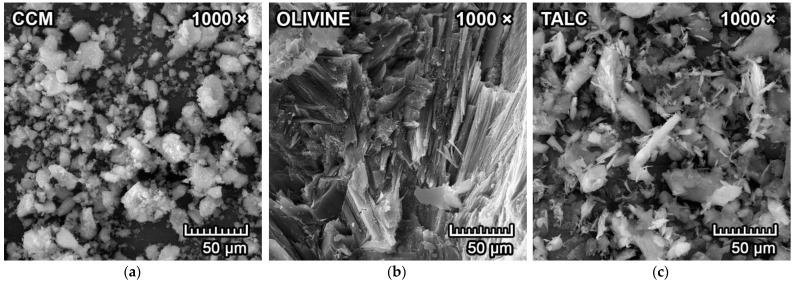
Scanning electron microscope images of (**a**) CCM; (**b**) olivine; (**c**) talc; (**d**) fly ash; (**e**) alumina; (**f**) kaolin with 1000× magnification.

**Figure 3 materials-15-01363-f003:**
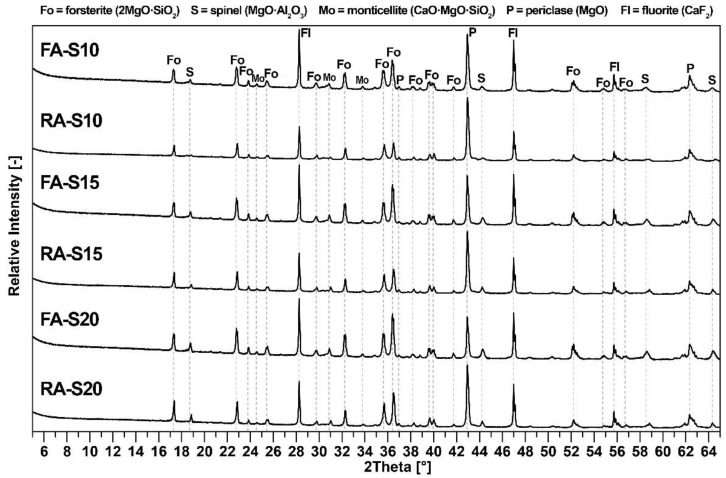
X-ray diffraction analysis of all fly ash–based and reactive alumina–based mixtures.

**Figure 4 materials-15-01363-f004:**
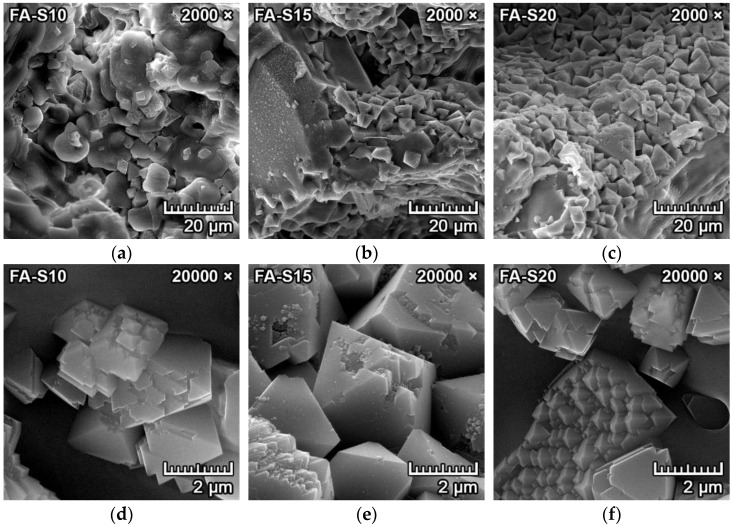
SEM images of fired forsterite–spinel ceramics mixtures with fly ash (FA-S10, S15, S20) with a magnification of 2000× (**a**–**c**) and 20,000× (**d**–**f**).

**Figure 5 materials-15-01363-f005:**
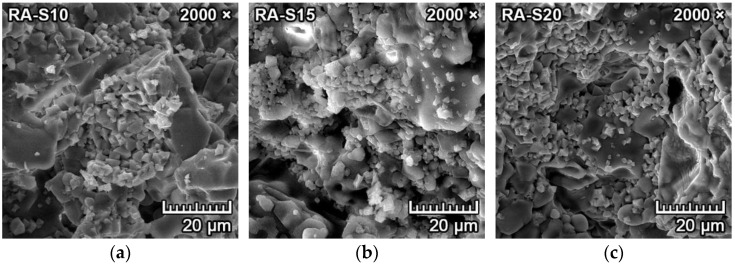
SEM images of fired forsterite–spinel ceramics mixtures with reactive alumina (RA-S10, S15, S20) with a magnification of 2000× (**a**–**c**) and 20,000× (**d**–**f**).

**Figure 6 materials-15-01363-f006:**
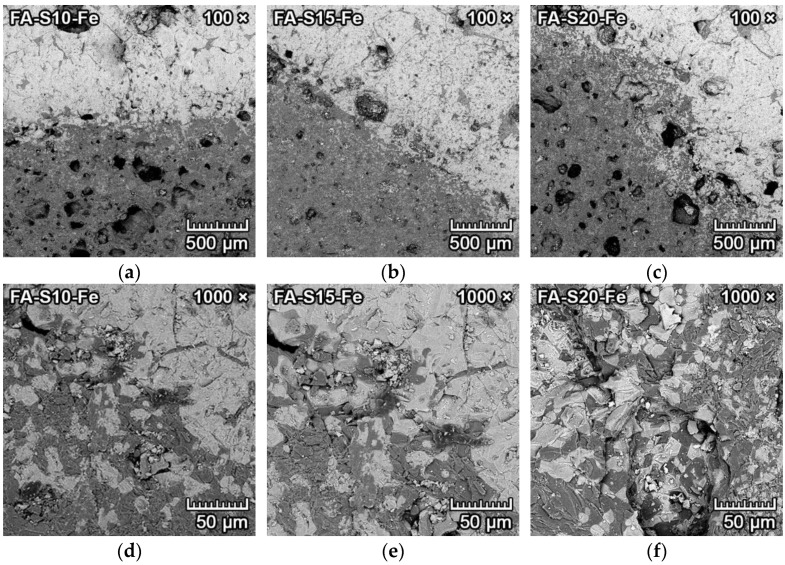
SEM microstructure of the transition zone between iron (light grey) and forsterite–spinel ceramics (dark grey) of fly ash–based mixtures FA-S10, FA-S15 and FA-S20, with a magnification of 100× (**a**–**c**) and a magnification of 1000× (**d**–**f**).

**Figure 7 materials-15-01363-f007:**
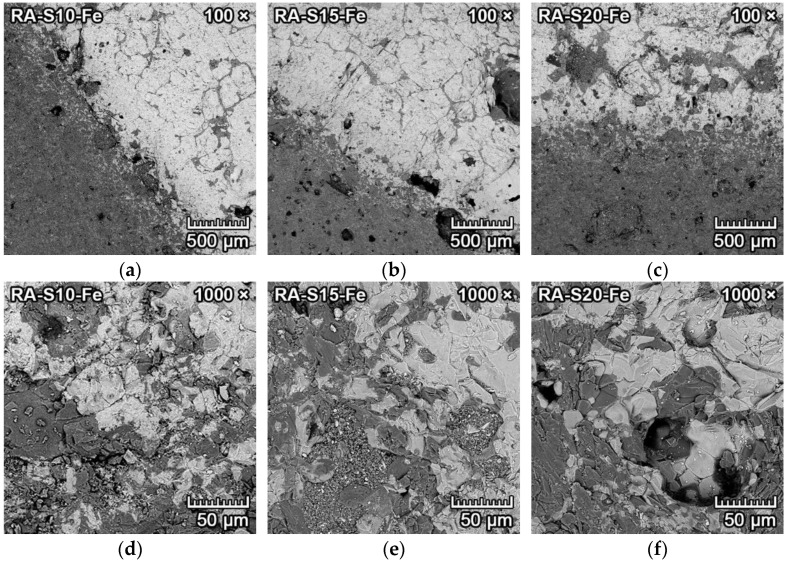
SEM microstructure of the transition zone between iron (light grey) and forsterite–spinel ceramics (dark grey) of reactive alumina–based mixtures RA-S10, RA-S15 and RA-S20, with a magnification of 100× (**a**–**c**) and a magnification of 1000× (**d**–**f**).

**Figure 8 materials-15-01363-f008:**
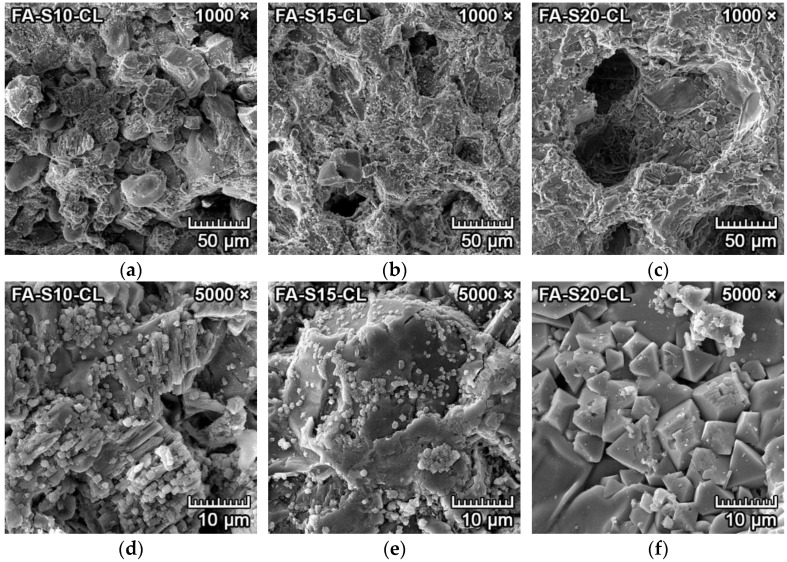
SEM microstructure of the transition zone between clinker and forsterite–spinel ceramics of fly ash–based mixtures FA-S10, FA-S15 and FA-S20, with a magnification of 1000× (**a**–**c**) and a magnification of 5000× (**d**–**f**).

**Figure 9 materials-15-01363-f009:**
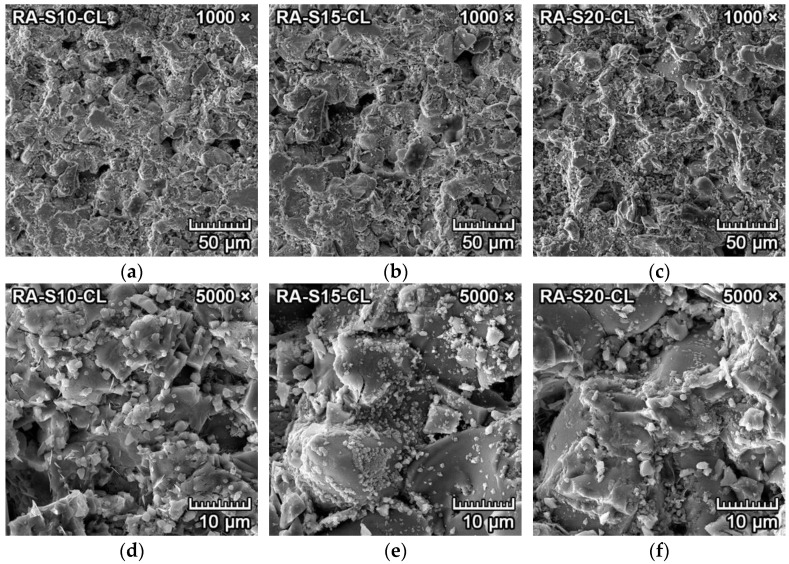
SEM microstructure of the transition zone between clinker and forsterite–spinel ceramics of reactive alumina–based mixtures RA-S10, RA-S15 and RA-S20, with a magnification of 1000× (**a**–**c**) and a magnification of 5000× (**d**–**f**).

**Figure 10 materials-15-01363-f010:**
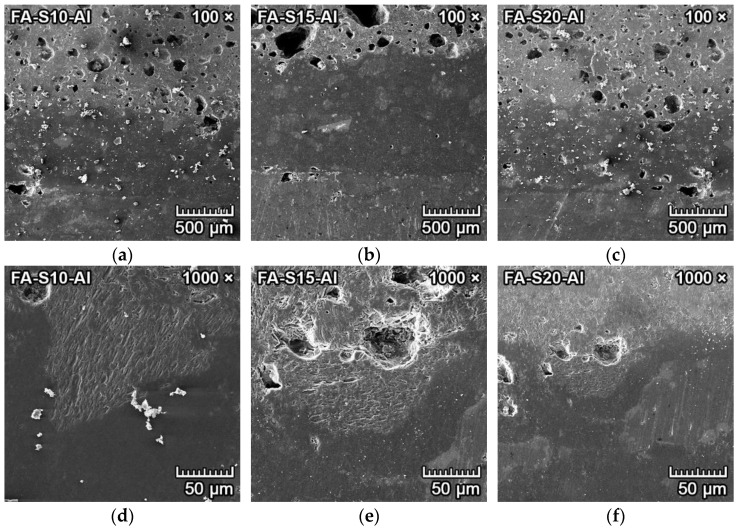
SEM microstructure of the transition zone between aluminum (dark grey and smooth) and forsterite–spinel ceramics (lighter and porous) of fly ash–based mixtures FA-S10, FA-S15 and FA-S20, with a magnification of 100× (**a**–**c**) and a magnification of 1000× (**d**–**f**).

**Figure 11 materials-15-01363-f011:**
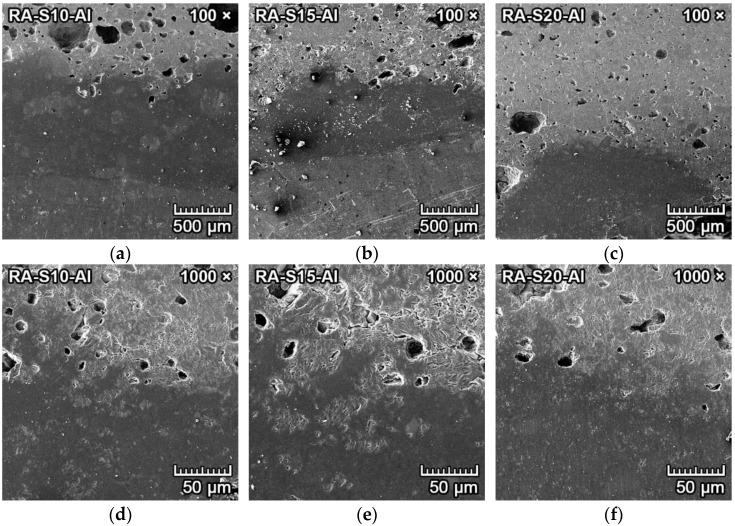
SEM microstructure of the transition zone between aluminum (dark grey and smooth) and forsterite–spinel ceramics (lighter and porous) of reactive alumina–based mixtures RA-S10, RA-S15 and RA-S20, with a magnification of 100× (**a**–**c**) and a magnification of 1000× (**d**–**f**).

**Figure 12 materials-15-01363-f012:**
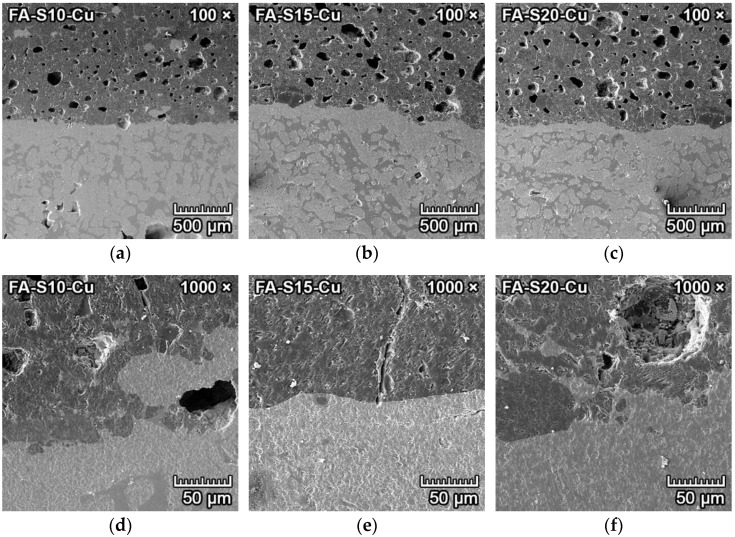
SEM microstructure of the transition zone between copper (light grey and smooth) and forsterite–spinel ceramics (darker and porous) of fly ash–based mixtures FA-S10, FA-S15 and FA-S20, with a magnification of 100× (**a**–**c**) and a magnification of 1000× (**d**–**f**).

**Figure 13 materials-15-01363-f013:**
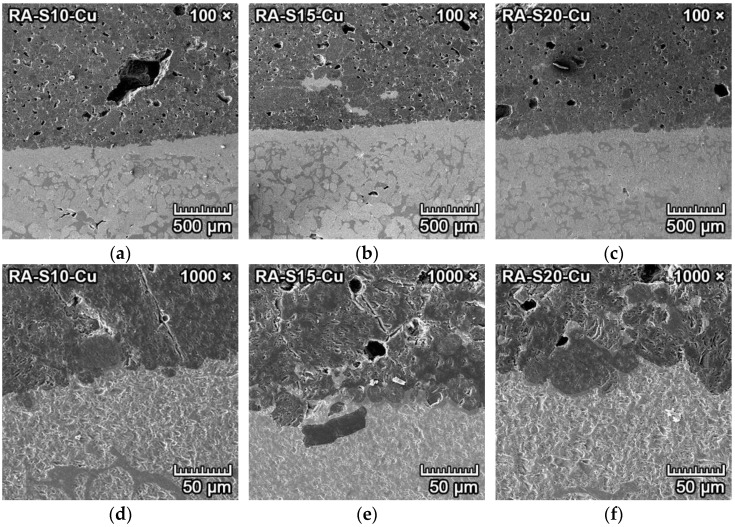
SEM microstructure of the transition zone between copper (light grey and smooth) and forsterite–spinel ceramics (darker and porous) of reactive alumina–based mixtures FA-S10, FA-S15 and FA-S20, with a magnification of 100× (**a**–**c**) and a magnification of 1000× (**d**–**f**).

**Table 1 materials-15-01363-t001:** The chemical composition of all used raw materials.

Raw Materials	MgO	SiO_2_	Al_2_O_3_	CaO	Fe_2_O_3_	K_2_O+Na_2_O	TiO_2_	LOI ^1^
CCM	85.6	0.5	0.1	5.2	7.4	0.1	0.1	1.0
Olivine	24.1	64.7	1.0	0.7	8.8	0.5	0.1	0.1
Talc	31.5	59.1	1.0	1.0	0.7	0.2	0.0	6.5
Fly ash	1.4	57.4	29.3	2.2	5.1	1.7	1.7	1.2
Alumina	0.0	0.0	99.3	0.0	0.1	0.3	0.0	0.3
Kaolin	0.5	46.8	36.6	0.7	0.9	1.2	0.1	13.2

^1^ Loss on ignition.

**Table 2 materials-15-01363-t002:** Composition of all designed mixtures from raw materials.

Raw Materials	FA-S10	FA-S15	FA-S20	RA-S10	RA-S15	RA-S20
CCM [wt.%]	43.5	45.9	48.2	40.0	39.6	39.2
Olivine [wt.%]	24.8	15.9	7.1	34.0	32.1	30.2
Talc [wt.%]	12.4	8.0	3.5	17.0	16.0	15.1
Fly ash [wt.%]	14.3	25.2	36.2	-	-	-
Alumina [wt.%]	-	-	-	4.1	7.3	10.5
Kaolin [wt.%]	5.0	5.0	5.0	5.0	5.0	5.0

**Table 3 materials-15-01363-t003:** Quantitative phase composition of all mixtures.

Phase	FA-S10	FA-S15	FA-S20	RA-S10	RA-S15	RA-S20
Forsterite	57.8	48.5	42.4	72.2	66.5	58.3
Spinel	11.1	14.2	19.7	9.8	14.7	19.8
Periclase	13.4	14.8	11.9	8.3	10.1	10.6
Monticellite	2.7	2.2	2.4	1.6	2.0	1.9
Amorphous phase	14.9	20.4	23.6	8.1	6.7	9.4

**Table 4 materials-15-01363-t004:** Results of experiments to determine physico-mechanical properties.

Mixture	Apparent Porosity (%)	Water Absorption (%)	Bulk Density (kg·m^−3^)	Modulus of Rupture (MPa)
FA-S10	24.2	14.7	2365	15.5
RA-S10	17.5	4.5	2745	16.0
FA-S15	21.8	11.3	2460	18.4
RA-S15	15.6	4.2	2735	22.6
FA-S20	16.3	8.6	2510	17.3
RA-S20	14.4	2.9	2750	19.1

**Table 5 materials-15-01363-t005:** Results of experiments to determine refractory and thermo-mechanical parameters.

Mixture	Firing Shrinkage (%)	Refractoriness (°C)	Refractoriness under Load T_0.5_ (°C)	Residual MOR (%)
FA-S10	5.9	1694	1593	21.3
RA-S10	6.2	1737	1664	23.0
FA-S15	8.5	1676	1561	28.6
RA-S15	6.9	1742	1678	35.3
FA-S20	11.3	1655	1532	22.5
RA-S20	7.4	1731	1645	26.7

## Data Availability

The data presented in this paper are available upon request from the corresponding author.

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
