# Peer review of "Corrosion Resistance of Novel Fly Ash-Based Forsterite-Spinel Refractory Ceramics"

_materials, 2022, doi:10.3390/ma15041363_

Round 1

Reviewer 1 Report

This article presents an interesting topic about the corrosion resistance of novel fly ash–based forsterite–spinel (Mg2SiO4-MgAl2O4) refractory ceramics to various corrosive media in comparison with reactive alumina–based ceramics. Data are interesting, but I have some comments:

1.- I suggest improving introduction with more information about corrosion in this kind of materials.

2.- In Table 1, the addition of chemical components CCM and Fly Ash are not 100%.

3.- In page 3, lines from 99 to 100, about Figure 1, X-Ray diffraction, is mentioned that the major crystalline phase in CCM was periclase (MgO), with trace amounts of iron compounds, however, the iron compounds are not included into the X-Ray pattern.

4.- In page 4, lines from 126 to 128 is mentioned: “After homogenization, mixtures were then mixed with a precisely determined amount of water, utilizing a Pfefferkorn apparatus (standard ÄŒSN 72 1074) to achieve optimal plasticity.”, What was the amount of water? What was the optimal plasticity?

5.- In page 8, Item “3.3.1. Corrosion resistance to iron”, What kind of iron was used?

6.- In page 11, lines from 319 to 320, Why apparent porosity and water absorption decreased with increased amounts of formed spinel in the structure in both fly ash mixtures and reactive alumina mixtures? Please widen the discussion.

Reviewer 2 Report

Review of the article

Manuscript Number: materials-1551363

Title:   Corrosion Resistance of Novel Fly Ash–Based Forsterite–Spinel Refractory Ceramics

In my opinion, the manuscript is not adequate for publishing in its current state.

In section 2.2. Sample Preparation authors wrote: In this work, six different mixtures were designed. Please explain, designed based on what?

In section 2.3. please elaborate investigation of corrosion resistance - which materials and why there are different temperatures for different corrosive media used and every time when corrosive resistance is mentioned – corrosion resistance of what? Please specify (all six mixtures or?).

SEM images of raw materials (olivine, alumina and coal fly ash) are presented in Figures 2a–c. Please provide SEM images of all raw materials (SEM images of CCM, talc and kaolin are missing). From Figure 4 to 8 please specify which image is for which refractory ceramics (of 6 prepared mixtures).

The section 4. Conclusions – from the written paragraphs it is not clear how and based on what and for which of the 6 mixtures you made these conclusions. It is not in correlation to the discussion and results.

All in all, it is not elaborated enough and it’s not comprehensive while reading.

Reviewer 3 Report

Interesting research. FTIR analysis is missing. See how the formation of aluminosilicate gel DOI is explained 10.3390 / gels7040195. See also this DOI research 10.2298 / NTRP201120006N. The results of XRD and SEM analysis would be complemented by FTIR analysis.

Round 2

Reviewer 2 Report

Authors have to separate the following section in framework of 2. Materials and methods: 2.4. Investigation of ceramics corrosion resistance and elaborate.    It's unclear whether all six mixtures are investigated in all four corrosive media from the manuscript - please specify every time. Why are there no SEM images for all six of prepared ceramic mixtures - why did you single out only two of them (FA-S15 and RA-S15)? There have to be results for all of them if you compare them in the text later OR explain why you selected only two - without complete results it is too confusing.  Also, there are no complete SEM microstructure images of prepared ceramics before exposing to corrosive media. Please specify throughout the text for which sample every result is - add complete results and information so that everything is clear. Please consider the correct order of writing chapters - the manuscript is disorganized and confusing. 
